# (E)-2-Cyano-3-(1*H*-Indol-3-yl)-N-Phenylacrylamide, a Hybrid Compound Derived from Indomethacin and Paracetamol: Design, Synthesis and Evaluation of the Anti-Inflammatory Potential

**DOI:** 10.3390/ijms21072591

**Published:** 2020-04-08

**Authors:** Pablo Silva, Maria de Almeida, Jamire Silva, Sonaly Albino, Renan Espírito-Santo, Maria Lima, Cristiane Villarreal, Ricardo Moura, Vanda Santos

**Affiliations:** 1Programa de Pós-Graduação em Ciências Farmacêuticas, Universidade Estadual da Paraíba, Campina Grande 58429-500, PB, Brazil; pablo-rayff@hotmail.com (P.S.); sonaly.albino@hotmail.com (S.A.); ricardo.olimpiodemoura@gmail.com (R.M.); 2Laboratório de Ensaios Farmacológicos, Departamento de Farmácia, Universidade Estadual da Paraíba, Campina Grande 58429-500, PB, Brazil; cris.freire21@hotmail.com; 3Laboratório de Desenvolvimento e Síntese de Fármacos, Departamento de Farmácia, Universidade Estadual da Paraiba, Campina Grande 58429-500, PB, Brazil; jamiremuriel@hotmail.com; 4Programa de Pós-Graduação em Ciências Farmacêuticas, Universidade Federal de Pernambuco, Recife 58429-500, PE, Brazil; nenalimamariadocarmo@gmail.com; 5Instituto Gonçalo Moniz, Fundação Oswaldo Cruz, Salvador 40296-710, BA, Brazil; r.fernandes88@hotmail.com (R.E.-S.); cfv@ufba.br (C.V.); 6Faculdade de Farmácia, Universidade Federal da Bahia, Salvador 40170-290, BA, Brazil

**Keywords:** cytokines, paw edema, PGE_2_, phenylacrylamide

## Abstract

The compound (E)-2-cyano-3-(1H-indol-3-yl)-N-phenylacrylamide (ICMD-01) was designed and developed based on the structures of clinically relevant drugs indomethacin and paracetamol through the molecular hybridization strategy. This derivative was obtained by an amidation reaction between substituted anilines and ethyl 2-cyanoacetate followed by a *Knoevenagel*-type condensation reaction with indole aldehyde that resulted in both a viable synthesis and satisfactory yield. In order to assess the immunomodulatory and anti-inflammatory activity, in vitro assays were performed in J774 macrophages, and significant inhibitions (*p* < 0.05) of the production of nitrite and the production of cytokines (IL-1β and TNFα) in noncytotoxic concentrations were observed. The anti-inflammatory effect was also studied via CFA-induced paw edema in vivo tests and zymosan-induced peritonitis. In the paw edema assay, ICMD01 (50 mg kg^−1^) showed satisfactory activity, as did the group treated with dexamethasone, reducing edema in 2–6 h. In addition, there was no significant inhibition of PGE_2_, IL-1β or TNFα in vivo. Moreover, in the peritonitis assay that assesses leukocyte migration, ICMD-01 exhibited promising results. Therefore, these preliminary studies demonstrate this compound to be a strong candidate for an anti-inflammatory drug together with an improved gastrointestinal safety profile when compared to the conventional anti-inflammatory drugs.

## 1. Introduction

According to the World Health Organization (WHO), diseases of inflammatory etiology constitute the greatest hazard to human life. They are the main causes of death in the world, and are estimated to become more prominent causes of death over the next 30 years in the United States. Rand Corporation studies in 2014 found that about 60% of the American population have at least one chronic inflammatory condition [1,2,3].

Pharmacological therapies used for the clinical treatment of inflammatory diseases include non-steroidal (NSAIDs) and steroidal (SAIDs) drugs. Steroidal drugs, represented by glucocorticoids, are considered the gold standard acting on immune modulation and inflammatory processes. However, they have serious side effects associated with metabolic and endocrine disorders when used for prolonged periods [4].

Typical NSAIDs are the most prescribed drugs, being used by around 30% of the world’s population every day [5]. Among the main drugs used in the management of inflammatory and musculoskeletal conditions, indomethacin stands out. Indomethacin is considered the most potent non-selective NSAID and shows the highest ulcerogenic potential of the entire class due to both the inhibition of cytoprotective prostaglandins and gastric hyperstimulation. Associated with gastropathies, this drug is also responsible for promoting the formation of free radicals that potentially damage the nephron membrane [6,7,8].

*Acetaminophen*, or *N*-acetyl-to-aminophenol (APAP), regardless of not being considered a typical NSAID due to its weak anti-inflammatory potential, is a widespread drug commonly used since 1955 for analgesic and antipyretic purposes. Its use began to be restricted due to the reported cases of hepatotoxicity which started to appear in the USA during the 1980s. Mortality rates were approximately 0.4% in patients with overdoses, reaching approximately 300 cases of death per year. The toxic intake dose responsible for triggering the renal effects was considered to be above 150 mg kg^−1^. Currently, there are reports of insufficiency and acute liver injuries by lower doses. Furthermore, it is expected that during the year, about 30,000 patients will be admitted to hospitals for the treatment of hepatotoxicity by APAP [9,10].

Such situations have triggered the need to develop new anti-inflammatory drugs with preserved efficacy and better safety profiles compared to conventional drugs. One of the approaches that has been used in drug planning is medicinal chemistry, which provides a diversity of carbonic structures with pharmacophoric fragments important for the design of several bioactive compounds [11,12,13]. For example, the phenylacrylamide derivatives, a class resulting from the molecular hybridization of the *indole-acetic* acid *derivative* (indomethacin) and *N*-phenylacetamide (*acetaminophen*).

In this context, the present study was carried out to develop a hybrid compound and establish its pharmacological properties in models of inflammation in vitro and in vivo. For this, the fragmented-based drug design (FBDD) and structure-based drug design (SBDD) planning strategy were used based on the structures of paracetamol, indomethacin and topographic regions of COX. The final molecule, (E)-2-cyano-3-(1H-indol-3-yl)-*N*-phenylacrylamide (ICMD-01) was obtained through a *Knovenagel*-type condensation reaction with indole aldehyde. Compounds of the phenylacrylamide class, despite having several pharmacological applications, are poorly investigated for inflammation. Therefore, it will describe herein the development of a hybrid inedited compound with anti-inflammatory properties. 

## 2. Results and Discussion

### 2.1. Designing a New Hybrid with Anti-Inflammatory Potential

The hybrid (E)-2-cyano-3-(1*H*-indol-3-yl)-*N*-phenylacrylamide, called ICMD-01, was designed to contain fragments of two commercially and clinically important drugs, paracetamol and indomethacin. A study where the compound ICMD-01 was anchored at the enzymatic site of cyclooxygenases *2* (4COX) to analyze the possible anti-inflammatory actions was carried out. Moreover, the co-crystallized ligand indomethacin was used as a comparison in order to provide information on the possibility of better binding energy with respect to its coupling analogue with the receptor [14].

In the first stage of the procedure, indomethacin was redocked at the active site of COX-2, presenting a free binding energy of −11.26 kcal mol^−1^, while the proposed compound ICMD-01 bonded to the isoform of COX-2, resulting in a free energy binding value of −10.11 kcal mol^−1^. Therefore, in addition to presenting an interaction value approximated to the reference compound, the structure of ICMD-01 showed interactions in common with indomethacin in the crystallized target. The following binding sites stand out for both compounds: Ser: 530, Tyr: 385, Met: 522, Val: 523, Val: 349, Leu: 384 and Ala: 527. The complex formed between the compound ICMD-01, and the target COX-2 was stabilized mainly by conventional hydrogen interactions; however, the slight increase in free energy value of ICMD-01, when compared to that of indomethacin, can be attributed to differences in the type of binding with the amino acid residues, the least number of interactions and/or the presence of an unfavorable interaction at the enzymatic site (Gly: 526, unfavorable donor-donor). Anchoring positions of ICMD-01 and indomethacin at the active site of COX-2 are shown in Figure 1A,B, respectively.

### 2.2. Synthesis and Structural Elucidation

To obtain ICMD-01, an amidation reaction was carried out between substituted anilines and ethyl 2-cyanoacetate, followed by a *Knoevenagel-type* condensation reaction with the indole aldehyde in molar equivalence. The reaction took place under basic catalysis to form the carbanion that subsequently suffered a nucleophilic attack from the carbonyl carbon of the indole aldehyde followed by the elimination of water with consequent formation of the carbon–carbon double bond. The route used (Scheme 1) was suitable, with a satisfactory yield of 62.3%.

The structural elucidation of ICMD-01 was performed by nuclear magnetic resonance spectroscopy (1H and 13C NMR), infrared (IR) spectrometry and mass spectrometry. Spectral data were consistent with the proposed structure. In the IR spectrum, absorption bands were observed at 1651 cm^−1^; that is characteristic of the carbonyl group stretching. In addition, bands in the region 2206 and 3275 cm^−1^ were attributed to the nitrile, amide and amine groups, respectively. All absorption bands corroborate with the literature [14].

In the ^1^H NMR spectra, singlets referring to the NH group of the amide could be detected with chemical shifts of 10.17 and 12.54 ppm for the indole NH. In addition, it was also possible to view a singlet at 8.62 ppm, indicative of condensation between the aldehyde and the methylene group, resulting in the formation of the vinyl group (C = CH). Regarding the aromatic hydrogens, those from indole ring are between 7.29 and 8.04 ppm, and the ones from phenylacetamide function are between 7.70 and 7.13 ppm. All signs were identified as doublets, triplets or multiplets.

In the ^13^C DEPT-Q NMR spectra, two signals were observed in the negative region at 161.98 and 139.04 ppm, corresponding respectively to the carbonyl carbon and to the quaternary carbon, indicating *Knoevenagel* condensation. Likewise, it was possible to detect signals at 143.2 ppm that are indicative of vinyl carbons, and at 119.08 ppm, which is a diagnostic signal of nitrile function. Finally, the ICMD-01 mass spectrum showed a molecular ion peak (m/z = 310.088) consistent with its mass when added to the molar mass of the sodium ion, thereby aligning with the other spectroscopic analyses. The spectra referring to the IV (Appendix A) ^1^H NMR (Appendix A), ^13^C NMR (Appendix A) and HRMS (Appendix A) analyzes of the ICMD-01 are presented in Appendix A

### 2.3. In Silico Pharmacokinetic Predictions

In order to identify the possible pharmacokinetic drawbacks of ICMD-01, ADMET predictions were realized in silico. According to Lipinski’s rules, the potential drug is orally active when it meets the following criteria: (a) molecular weight ≤ 500 da; (b) LogP ≤ 5 (or MLogP ≤ 4.15); (c) number of hydrogen bond acceptors ≤ 10; (d) number of hydrogen bond donors ≤ 5 [15]. ICMD-01 has an MLogP of 1.99 and adequate numbers (based on the Lipinski rule) of donors and acceptors of hydrogen bonds which suggest a good oral availability (Table 1). Regarding LogS, the values found for the compound are out of the range (−4 and −2) which indicates moderate water solubility.

Parameters such as TPSA (TPSA ≤ 140 Å), logarithm of the apparent permeability coefficient (log Papp, log cm.s^−1^ > 0.9), were analyzed. Those indicate a high probability of ICMD-01 absorption in the human small intestine, with values >90%. Regarding the volume of distribution (VDss) that indicates the volume of liquid needed to contain all the drug in the organism in the same concentration present in the plasma, ICMD-01 presented intermediate values and must have an adequate plasma distribution profile demonstrating that the compound can be well distributed and readily available to interact with the target [16]. Lastly, it is possible to observe the clearance rate value of 0.318, slightly elevated, which may lead to a possible change in bioavailability. It is important to note that logP and the number of hydrogen donors are the main properties involved in the gain of oral bioavailability. The other factors, although of great importance, constantly undergo significant changes in the established parameters [17].

### 2.4. Biological Activity

The pharmacological potential of the derived phenylacrylamide ICMD-01 was investigated by a series of in vitro assays in J774 murine macrophage cells. Macrophages are important cells in the immune system performing various functions in the immune and adaptive response, systemic metabolism, hematopoiesis, angiogenesis, malignancy and reproduction. Regarding the inflammatory process control, these cells contribute to both homeostasis and the progression of tissue damage, modulating several inflammatory mediators [18,19,20,21].

The cell viability test was the first test performed that allowed for determining which concentrations would not cause cell death by the compound, and thus, measure its modulatory effects. As seen in Figure 2A, ICMD-01 showed no evidence of cytotoxicity in J774 cells at concentrations up to 100 μM which was, therefore, established as the maximum concentration for subsequent tests.

To stimulate inflammatory mediators such as cytokines and nitric oxide, macrophages are exposed to LPS and IFN-γ which act by modifying their intracellular balance [22]. This happens due to the activation of standard surface receptors of the Toll family (TLR), which activates several cellular signaling processes triggered by cytosolic adapter proteins, such as myeloid differentiation factor 88 (MyD88) and/or IFN-γ [Trif] domain inducer Toll/IL-1R, resulting instantly in the deflagration of acute inflammatory responses in macrophages [23,24].

Nitric oxide, a mediator widely expressed by activated macrophages, can be measured from nitrite quantification. Figure 2B shows that the treatment with ICMD-01 reduced the production of nitrite, suggesting a reduction in the production of NO. This inhibitory effect of ICMD-01 on J774 macrophages was statistically significant at concentrations of 100 and 50 μM (*p* < 0.05), and dexamethasone at a concentration of 40 μM was used as the standard in the test.

NO is a gaseous molecule synthesized by NOs which uses L-arginine, NADPH and oxygen as a substrate. Although there are three distinct enzyme isoforms, only inducible nitric oxide synthase (iNOs) is activated by microbial products in inflammatory cells such as macrophages. This biomolecule has different functions in the physiological and pathological responses in the body. Its modulation by ICMD-01 in inflammatory conditions may restrict immune responses involved in the defense of the host decreasing cell damage induced by the production of reactive oxygen species [25,26,27].

Concerning pro-inflammatory cytokines, also expressed by the activation of macrophages, treatment with ICMD-01 in concentrations of 100 and 50 μM was able to substantially suppress the release of TNF-α and IL-1β (*p* < 0.05) (Figure 3A,B). Statistically, under the same conditions, dexamethasone similarly inhibited the production of TNF-α and IL-1β when compared to the group treated with ICMD-01 at a concentration of 100 μM. The reduction in cytokines release by the test compound may act by limiting the inflammatory process due to the decreased activation of the NF-κB pathway, which acts as a transcriptional factor that plays a major role in the induction of inflammatory genes of pro-inflammatory molecules [28,29].

In vitro studies with a synthetic chalcone (6a) of the phenylacrylamide class disclosed that the compound contributed to decreased production of nitrite and pro-inflammatory cytokines in mesangial cells induced by LPS. The suppressor mechanism involved the activation of the factor Nrf2 (*nuclear factor erythhroid 2 related factor 2*), which seems to be one of the main ones involved in the expression of antioxidant proteins modulating inflammatory and immunological responses [30].

Nrf2-dependent antioxidant genes, such as *heme-oxygenase-1* (OH-1) and *NADP dehydrogenase quinone* (NQO-1), among others, act by blocking TNF-α—in addition to several monocyte chemotactic proteins and macrophage inflammatory proteins—thereby suppressing several inflammatory mediators as a result of reduced exposure to oxidative stress in target cells; and thus reducing the overproduction of cytokines that would activate the NF-κB pathway [31]. Activation of Nrf2 also prevents transcriptional positive regulation of LPS from cytokines such as TNF-α and IL-1β [32,33].

Therefore, it can be inferred that ICMD-01, due to its acrylate function that acts as a Michael acceptor, can react with Keap-1 cysteine groups, suppressing ubiquitination of Nrf2 and antioxidant enzyme expression [34]. Indirectly, this mechanism can promote nitrite and cytokine (IL-1β e TNFα) reduction in macrophage cultures to non-cytotoxic concentrations. Due to in vitro data which suggested the anti-inflammatory potential of ICMD-01, in vivo studies were carried out in order to assess the interference of biological systems and possible changes in pharmacological activity using paw inflammation models induced by Freund’s complete adjuvant (CFA). CFA is a complex consisting of a suspension of *Mycobacterium tuberculosis* and mineral oil which guarantees both deposition action and formation of granulomas rich in macrophages and immunocompetent cells [35,36,37].

Paw edema in the group treated with the dose of 25 mg kg^−1^ presented percentages of inhibition at 2, 4 and 6 h of 30.6%, 60.0% and 28.6%. The group treated with the dose of 50 mg kg^−1^ inhibited by 44.5%, 57.1% and 67.34%, and the group treated with 100 mg kg^−1^ by 55.6%, 54.3% and 51% when compared to the control group. The two highest doses show similar activity profiles among the three times, with no statistical difference, reducing edema in 2 to 6 h. These results establish the therapeutic dose for this compound at 50 mg kg^−1^, corresponding to its maximum effect. Furthermore, the group treated with ICMD-01 showed results comparable to the group treated with dexamethasone which presented an average inhibition of 62% (Figure 4).

The inflammatory process initiated by the phlogistic agent involves both fenestration of microvessels and leakage of liquid and cells to the interstitial tissue. The triggered cellular response induces the exacerbated release of a series of inflammatory mediators, such as cytokines, vasoactive amines, NO, chemotactic factors, interferons and prostaglandins [38]. The results found for the paw edema model suggest that the activity of ICMD-01 may partially be related to inhibition of mediators, such as IL-1β, TNFα and NO, as shown in the in vitro results.

Additionally, to evaluate the anti-inflammatory effect of ICMD-01 (50 mg kg^−1^*,* orally against leukocyte migration, a model of zymosan-induced peritonitis was performed. The zymosan used as a phlogistic agent is constituted by protein–carbohydrate complex extracted from the cell wall of the yeast *Saccharomyces cerevisiae*. This agent is mainly composed of β-glucans in addition to a non-uniform mixture of mannans, chitin, lipids and various proteins [39].

The biochemical basis of inflammatory response triggered by *zymosan* is multifactorial due to the complex mixture of cell wall components of the fungi. However, it is known that β-glucan activates the dectin-1 receptor and the Toll-2 type receptor (TLR2) primarily, initiating pathophysiological responses by DAMPs released following tissue damage [40,41,42]. Extravasated plasma has first-line chemotactic factors that mediate the subsequent influx of leukocytes, causing their accumulation in the inflamed tissue.

As presented in Figure 5, ICMD-01 inhibited polymorphonuclear migration by 63.2%, showing superior activity when compared to the group treated with indomethacin (10 mg kg^−1^, PO) which showed an inhibition of 45.3% when compared to the control group.

Leukocyte migration is a dynamic process, strictly regulated by chemokines, cytokines and adhesion molecules of the selectin family. These components promote the cellular rolling process of polymorphonuclear especially neutrophils at the injury site. The compound ICMD-01 can act by minimizing diapedesis by several direct or indirect mechanisms which act on either the adhesion system or on the expression of ICAM-1 (intercellular adhesion molecule), L-selectin (Leukocyte endothelial cell adhesion molecule, LECAM; CD62L), E-selectin (Endothelial leukocyte adhesion molecule-1, ELAM-1; CD62E) and P-selectin (CD62P).

Leukocytes, when bound to selectins, slow the free flow movement along the endothelium. Adhesion molecules are largely induced by cytokines and nitric oxide produced by macrophages and resident cells. Therefore, the potential of the compound in vitro against the inhibition of NO, TNF-α and IL-1β may justify the inhibition of leukocyte migration.

To validate the anti-inflammatory profile, the modulating potential of ICMD-01 (50 mg kg^−1^, PO) against the production of pro-inflammatory cytokines and prostaglandin E_2_ (PGE_2_) was measured in vivo. Cytokines are a diverse group of mediators widely released during inflammation. The compound ICMD-01 showed statistically significant inhibition (*p* < 0.05) of IL-1β and TNF-α local levels (Figure 6A,B) when compared to the control group. IL-1β inhibition promoted by ICMD-01 is comparable to that of dexamethasone, while for TNF-α the compound proved to be superior.

IL-1β is associated with increased excitability and sensitization of nociceptive endings, in addition to inducing apoptosis processes during inflammatory and carcinogenic responses. TNF-α constitutes a pleiotropic cytokine which induces the production of other cytokines, such as IL-1 and IL-6, promoting chemotaxis, cell recruitment, increased vascular permeability and hyperalgesia [43,44].

Due to the anti-inflammatory action of NSAIDs and their actions on cyclooxygenase indicated by the in silico study, the effect of ICMD-01 on local levels of PGE_2_ during inflammation caused by CFA was evaluated. As shown by the analysis, ICMD-01 (50 mg kg^−1^) inhibited PGE_2_ levels in the inflamed paw (*p* < 0.001) when compared to the control group, like the results from the group treated with dexamethasone (2 mg kg^−1^) (Figure 7).

After inflammatory stimulation, the extensive release of cytokines promotes the transcription of COX-2 due to the activation of the NFκB factor [45]. COX-2 is the enzyme responsible for the peroxidation reactions that will generate prostaglandins [46,47]. PGE_2_ is one of the main mediators involved in the appearance of the inflammatory process signs and symptoms, causing increased vascular permeability, vasodilation and edema, in addition to increasing the sensitivity of nociceptive endings, causing pain [48,49].

As a result of PGE_2_ suppression by ICMD-01, the possibility of the compound presenting induction of peptic ulcers as side effects was evaluated. These events are largely induced by non-selective anti-inflammatory drugs which act by suppressing the production of cytoprotective mucosal prostaglandins by COX-1. However, there was no statistical difference between the group treated with ICMD-01 and the control group (Figure 8), indicating low ulcerogenic potential when used for a period corresponding to 7 days in the therapeutic dose of 50 mg kg^−1^.

## 3. Materials and Methods 

### 3.1. Chemistry 

All reagents used in this study were commercially available Sigma-Aldrich (St. Louis, MO, USA), Acros Organic (Morris Plains, NJ, USA), Vetec (Rio de Janeiro, Brazil). The melting points were determined on Quimis 431D (Quimis, São Paulo, Brazil) apparatus and were uncorrected. IR spectra were measured on Shimadzu IRPrestige-21 spectrophotometer (Shimadzu, Kyoto, Japan) by the attenuated total reflection technique (ATR). NMR spectra were recorded on Bruker Avance Spectrometer AC (500 MHz for ^1^H and 125 MHz for ^13^C) instruments by using tetramethylsilane as an internal standard. DMSO-*d*_6_ was purchased from Sigma-Aldrich. The chemical shifts were reported in δ units and coupling constants (*J*) were reported in Hertz (Hz). The multiplicities were given as s (singlet), d (doublet), t (triplet), m (multiplet), dd (double doublet). The exact mass were obtained on Spectrometer MALDI-TOF Autoflex III (Bruker Daltonics, Billerica, MA, USA). TLC development was conducted on 0.25 mm silica gel plates (Merck, silica gel 60 F_254_ in aluminum foil).

### 3.2. Procedure for the Synthesis of Compound (E)-2-Cyano-3-(1H-Indol-3-yl)-N-Phenylacrylamide -ICMD-01

To begin, 2-cyano-*N*-phenylacetamide (1 mmol) was added to a solution of índole-3-carboxaldeyde (1 mmol), and 5–10 drops of triethylamine was used as the catalyst in toluene (10 mL). The reaction was processed under magnetic stirring for 96h with temperature controlled at 110 °C. The precipitate was removed under filtration and washed with distilled water, and then dried in a desiccator under vacuum. After drying, the products were recrystallized from ethanol. 

### 3.3. Physical-Chemical Properties and Spectroscopic Data of (E)-2-Cyano-3-(1H-Indol-3-yl) -N-Phenylacrylamide -ICMD-01

ICMD-01 was obtained as yellow powder (62.33%). M.p. (°C): 273–275. Rf (7:3 AcOEt/*n*-hexane): 0.45. IR (ATR): 1227 (NH, phenylacetamide), 1651 (C=O), 2206 (CN), 3275 (NH), 3060 (=C-H), 1575–1435 (C=C, Ar) cm^−1^. NMR ^1^H (500 MHz, DMSO*d_6_*): δ 12.43 (1H, s, NH indole); 10.17 (1H, s, NH amide); 8.62 (1H, s, C=CH); 8.55 (1H, s, C=CH indole); 8.04–7.98 (1H, m, Ar indole); 7.70 (2H, d, *J* = 7.9, Ar phenylacetamide); 7.59 (1H, m, Ar indole); 7.37 (2H, t, *J* = 7.8, Ar phenylacetamide); 7.29 (2H, pd, Ar indole); 7.13 (1H, t, *J* = 7,4, Ar phenylacetamide). NMR de ^13^C (124 MHz, DMSO*d_6_*): δ 161.98 (C, C=O); 143.2 (C, C=C); 140.4 (C, N-phenylacetamide); 137.0(C, Ar indole); 131.1 (C, Ar phenylacetamide); 129.1 (C, Ar indole); 128.1 (C, Ar); 124.4 (C, Ar indole); 123.8 (C, Ar indole); 122.1 (C, Ar); 121.1(C, Ar indole); 119.1 (C, CN); 113.2 (C, Ar indole); 110.3 (C, adjacent to CN). HRMS *m*/*z* [M + Na] ^+^ calculated for C_18_ H_13_ N_3_ONa: 310.11; found: 310.08.

### 3.4. In Silico Studies

#### 3.4.1. In Silico ADME and Molecular Docking Analysis

*In silico* molecular interaction, analysis studies on the test compound with COX-2 were performed using AutoDock4.2.6 in combination with the Lamarckian genetic algorithm. The ADME profiles of the synthetized compounds were investigated with the use of SwissADME [50] and pkCSM [51] web platforms.

#### 3.4.2. Molecular Docking

##### Ligand Structure Preparation

The structure of the test compound was built with Chemdraw professional 3D 15.0 software and fully optimized in semiempirical method MM2. The optimized ligands were then saved as Mol2. With the use of AutoDockTools-1.5.6, non-polar hydrogens were merged with the corresponding carbons, and then partial charges of atoms were calculated using the Gasteiger procedure implemented in the AutoDockTools package. Finally, the rotatable bonds of the ligands were defined, and the structures were saved as pdbqt and used for docking studies.

##### Protein Structure Preparation 

The crystal structure of indomethacin co-crystallized with COX-2 (PDB ID: 4COX) was retrieved from RCSB Protein Data Bank as described [52,53]. With the use of Molecular Graphics System, PyMOL, water molecules and other heteroatoms were removed. Then, using AutoDockTools, non-polar hydrogens were merged, and polar hydrogens added to the structures of the proteins. Kollman charges were added and the structure was saved as pdbqt for the docking studies.

##### Docking Procedure

The 3D grid was created by the Autogrid Algorithm to generate the grid parameter file. The grid spacing was 0.0375 nm in each dimension, and each grid map consisted of a 70 × 70 × 70 grid point. The center of the grid was set to the position of the co-crystalized ligand located at the Chain A. The Lamarckian genetic algorithm in AutoDock 4.2.6 was applied to search the best conformation and orientation of the ligands. The global optimization was started with a population of 150 randomly positioned individuals with a maximum of 2,500,000 energy evaluations and a maximum of 27,000 generations. During each docking experiment, 100 runs were carried out. The resulting docking poses were analyzed using AutoDockTools and Discovery Studio Visualizer. To validate the docking procedure, the co-crystalized ligand indomethacin was previously docked to the protein, obtaining a RMSD value of 0.91 Ǻ in the redocking procedure (Appendix A). In this case, a 70 × 70 × 70 grid map was used, and 100 runs were carried out.

### 3.5. Biological Activity

#### 3.5.1. Chemicals and Drugs

Dexamethasone, complete Freund’s adjuvant (CFA), phosphate buffered saline (PBS), dimethylsulfoxide (DMSO), phenylmethylsulphonyl fluoride (PMSF), benzamethonium chloride, EDTA, aprotinin A, Dulbecco’s modified Eagle’s medium (DMEM) and 3,3´,5,5´- tetramethylbenzidine (TMB) were obtained from Sigma Chemical Company (St. Louis, MO, USA. Dexamethasone was dissolved in ethanol (10% in normal saline solution). Indomethacin was dissolved in Tris HCl 0.1 M pH 8.0 plus saline solution. ICMD-01 was dissolved in 5% DMSO plus saline, and remaining substances were dissolved directly in saline. ICMD-01 was administered orally and the other drugs were administrated by intraperitoneal (i.p.) route 40 min before testing, and the control group only received vehicle.

#### 3.5.2. Cytotoxicity to Mammalian Cells

To determine the cytotoxicity of ICMD-01, the murine macrophage-like cell line J774 was plated into 96-well plates at a cell density of 2 × 10^5^ cells/well in Dulbecco’s modified Eagle medium (DMEM; Life Technologies, GIBCO-BRL, Gaithersburg, MD, USA) supplemented with 10% fetal bovine serum (FBS; GIBCO) and 50 µg/mL of gentamycin (Novafarma, Anápolis, GO, Brazil), and incubated for 2 h at 37 °C and 5% CO_2_. After that time, ICMD-01 was added at five concentrations ranging from 6.25 to 100 µM and incubated for 24 h. Twenty microliters per well of Alamar Blue (Invitrogen; Carlsbad, CA, USA) was added to the plates during 12 h. Colorimetric readings were performed at 570 and 600 nm. Gentian violet (Synth, São Paulo, Brazil) at 10 μM was used as positive control [52]. 

#### 3.5.3. Assessment of Cytokine and Nitric Oxide Production by Macrophages

For the evaluation of cytokine and nitric oxide production, J774 cells were seeded in 96-well tissue culture plates at 2 × 10^5^ cells/well in DMEM medium supplemented with 10% FBS and 50 µg/mL of gentamycin for 2 h at 37 °C and 5% CO_2_. Cells were then stimulated with LPS (500 ng/mL, Sigma Chemical, St. Louis, USA) and IFN-γ (5 ng/mL; Sigma Chemical, St. Louis, MO, USA) in the presence of ICMD-01, vehicle or dexamethasone at different concentrations, and incubated at 37 °C. Cell-free supernatants were collected 4 h (for TNF-α measurement) and 24 h (for IL-1β and nitrite quantification) and kept at −80 °C. Cytokine concentrations in supernatants from J774 cultures were determined by enzyme-linked immunosorbent assay (ELISA), using the DuoSet kit from R&D Systems, according to the manufacturer’s instructions. Quantification of nitric oxide was estimated by assessing the nitrite concentrations using the Griess method [53]. 

#### 3.5.4. Animals

Experiments were performed on male Swiss Webster mice obtained from the Animal Facilities at the Instituto Gonçalo Moniz (FIOCRUZ; Salvador, Brazil). Animals (22–28 g) were housed in temperature-controlled rooms (22–25 °C), under a 12:12 h light–dark cycle, with access to water and food ad libitum until experimental initiation. All behavioral tests were performed between 8:00 a.m. and 5:00 p.m., and animals were only used once. Animal care and handling procedures were in strict accordance with the recommendations in the Guide for the Care and Use of Laboratory Animals of the National Institutes of Health and Brazilian College of Animal Experimentation. The protocol was approved by Committee, Ethics Committee for Animal Experimentation of the Universidade Federal da Bahia (CEUA/ICS; 22 Feb 2019). Permit Number: 135/2018, validity 01 Feb 2023. Every effort was made to minimize the number of animals used and any discomfort. Behavioral tests were performed without knowing to which experimental group each mouse belonged. Results shown are from two independent experiments performed.

#### 3.5.5. Inflammatory Model 

Mice were lightly anesthetized with halothane and received 20 μL of complete Freund’s adjuvant (CFA 1 mg/mL of heat killed *Mycobacterium tuberculosis* in 85% paraffin oil and 15% mannidemonoleate; Sigma) in the plantar region of the right hind paw, according to a previously reported method [54]. Paw edema, local cytokines, and prostaglandin E_2_ levels were measured by plesthismometer and ELISA, respectively, as described below. Mice were injected with ICMD-01 (orally, 5, 50 and 100 mg kg^−1^), vehicle (p.o, 5% DMSO in Physiological saline; control group) or dexamethasone (i.p, 2 mg kg^−1^, reference drug) route 40 min before CFA.

##### Plesthismometer Test

The volume of each mouse paw was measured (mm^3^) with a plesthismometer (Ugo Basile, Comerio, Italy) before (Vo) and after (VT) the CFA injection, as described previously [54]. The amount of paw swelling was determined for each mouse and data were represented as paw volume variation (Δ, mm^3^).

##### Cytokine Measurement by ELISA

The paw levels of cytokines were determined as previously described [54]. Treatments were performed 40 min before the CFA injection. Skin tissues were removed from the paws 4 h or 24 h after CFA, in mice terminally anesthetized with halothane from each experimental group. Tissue proteins were extracted from 100 mg tissue/mL phosphate buffered saline (PBS) to which 0.4 M NaCl, 0.05% Tween 20 and protease inhibitors (0.1 mM PMSF, 0.1 mM benzethonium chloride, 10 mM EDTA, and 20 KI aprotinin A/100 mL) were added (Sigma Chemical Company, St. Louis, MO, USA). The samples were centrifuged for 10 min at 3000 g and the supernatants were frozen at −70 °C for later quantification. TNF-α and IL-1β levels were estimated using commercially available immunoassay ELISA kits for mice (R&D System, Minneapolis, MN, USA), according to the manufacturer’s instructions. The results are expressed as picograms of cytokine per milligram of protein.

##### Measurement of PGE_2_ in Paw Skin 

The plantar tissues were collected 3 h after intraplantar injection of CFA (20 µL/paw). The paws were injected with indomethacin (50 μg/paw) 10 min before tissue retrieval to block PGE_2_ production during tissue processing. The PGE_2_ levels were determined by radioimmunoassay, as previously described [55]. Briefly, the plantar tissue samples were homogenized in a mixture of 3.0 mL of extraction solvent (isopropanol/ethyl acetate/0.1 N HCl, 3:3:1) and 3.0 mL of distilled water containing 20 μg/mL of indomethacin. Homogenates were centrifuged at 1500× *g* for 10 min at 4 °C. The organic phase was aspirated and evaporated to dryness in a centrifugal evaporator. The pellet was reconstituted in 500 μL of 0.1 M phosphate buffer (pH 7.4) containing 0.8% sodium azide and 0.1% gelatin. Concentration of PGE_2_ in these samples was then measured by RIA by using a commercially available kit. The results are expressed as picograms of PGE_2_ per milligram of protein.

##### Zymosan-Induced Acute Peritonitis in Mice

The peritonitis was determined as previously described [56,57]. The mice were treated (p.o.) with 5% DMSO plus saline, 10 mg kg^−1^ indomethacin and ICMD-01 at 50 mg kg^−1^. After 1 h of the treatment, the animals were intraperitoneally injected with 2% zymosan. After 4 h, the animals were euthanized, and 3 mL of heparinized PBS was injected into the peritoneal cavity. The cells were resuspended in 500 μL PBS and an aliquot of 10 μL was diluted with Turk’s solution (1:20). The total leukocytes were counted in a Neubauer chamber using a light microscope, examining four external quadrants.

##### Ulcerogenic Liability Study

The ICMD-01 has been tested for its ulcerogenic liability using the previously reported method with adaptations [58]. The control group received the vehicle (5% DMSO plus saline) while the other group received the test compound at a dose of 50 mg kg^−1^; then animals were fed after 2 h. Mice were given the specified fosse orally for seven successive days. Mice were euthanized 2 h after the last dose; then the stomach of each mouse was removed and opened along the greater curvature for determination of the ulcer number and ulcer index. 

### 3.6. Statistical Analysis 

Data are presented as means ± standard errors of the means (SEM) of measurements made on 6–9 animals in each group. Comparisons between three or more treatments were made using one-way ANOVA with Tukey’s post-hoc test, or for repeated measures, two-way ANOVA with Bonferroni’s post-hoc test, as appropriate. All data were analyzed using Prism 5 Computer Software (GraphPad, San Diego, CA, USA). Statistical differences were considered to be significant at *p* < 0.05. 

## 4. Conclusions

Altogether, this study demonstrates for the first time, the in vivo anti-inflammatory properties of the phenylacrylamide derivative ICMD-01. Results suggest that the anti-inflammatory effects of this new compound may partially be related to the suppression of mediators, such as cytokines, nitric oxide and prostaglandin. This preliminary study supports ICMD-01 as a strong anti-inflammatory drug candidate that exhibits potentiated pharmacological responses and slighter undesirable effects related to gastrointestinal damage associated with the prolonged use of conventional drugs.

## 5. Patents

Privilege of Innovation. Registration number: BR10201902032; title: Hybrid Compounds (E)-2-Cyano-3-(1*H*-indol-3-yl) N-phenylacrylamide with immunomodulatory, anti-inflammatory and analgesic potential. Registration institution: INPI—Instituto Nacional da Propriedade Industrial. Deposit: 29/09/2019.

## Figures and Tables

**Figure 1 ijms-21-02591-f001:**
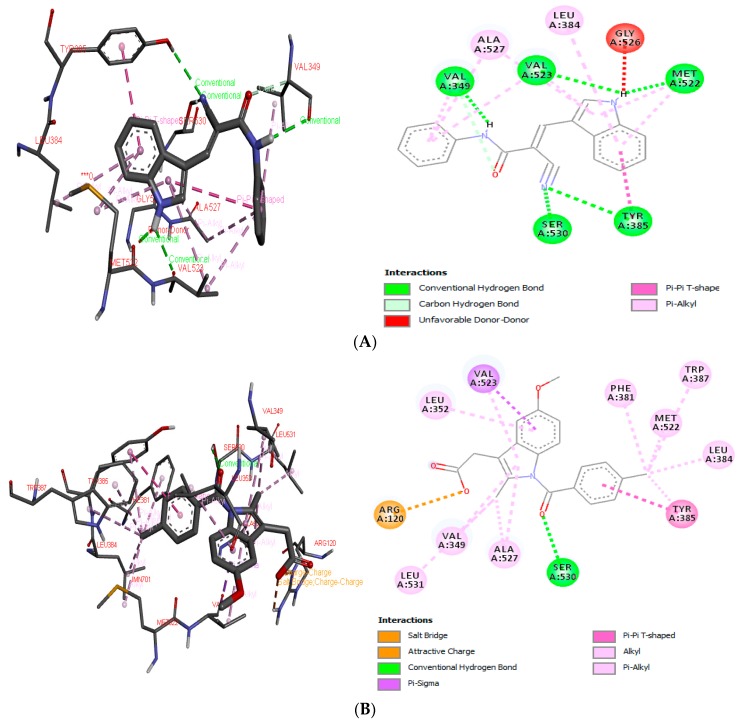
3D and 2D interaction diagram of ICMD-01 (**A**) and reference drug indomethacin (**B**) with active site of target receptor human cyclooxygenase-2 (4COX).

**Scheme 1 ijms-21-02591-sch001:**
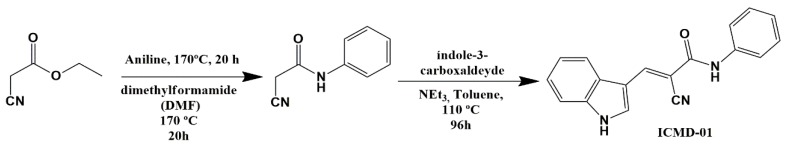
Synthesis of (E)-2-cyano-3-(1H-indol-3-yl)-*N*-phenylacrylamide—ICMD-01.

**Figure 2 ijms-21-02591-f002:**
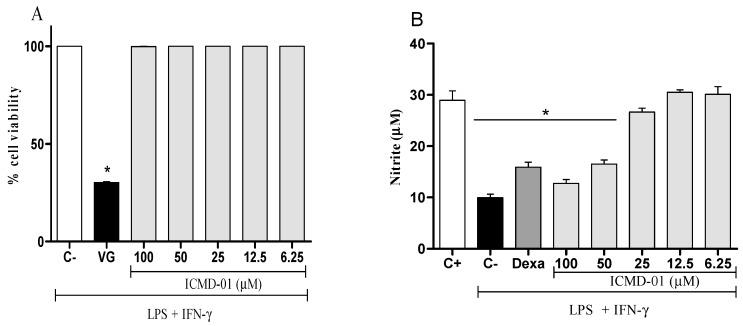
Cytotoxic effect of ICMD-01 (**A**) and its modulation of nitric oxide production on macrophages (**B**). Panel (**A**): J774 cells were incubated with vehicle (5% DMSO in saline, control group) or different concentrations of ICMD-01 (ICMD-01; 6.25, 12.5, 25, 50 or 100 μM) for 72 h, and cell viability was determined by Alamar Blue assay. Gentian violet (GV) was used as the positive control. Data are expressed as means ± SEMs; *n* = 4 determinations per group. * Significantly different from the vehicle treated cultures (*p* < 0.05). ANOVA followed by Tukey’s multiple comparison test. Panel (**B**): Concentrations of nitrite were determined in J774 macrophages treated with vehicle (5% DMSO in saline, control group), IMCD-01 (ICMD-01; 6.25, 12.5, 25, 50 or 100 μM) or dexamethasone (Dexa; 40 μM) in the presence of LPS (500 ng/mL) plus IFN-γ (5 ng/mL). Cell-free supernatants were collected 24 h after treatments for nitrite quantification by the Griess method. Ct- shows concentrations of nitrite in unstimulated cells. Data are expressed as means ± SEMs; *n* = 6 determinations per group. * Significantly different from the vehicle treated cultures stimulated with LPS + IFN-γ (*p* < 0.05). ANOVA followed by Tukey´s multiple comparison test.

**Figure 3 ijms-21-02591-f003:**
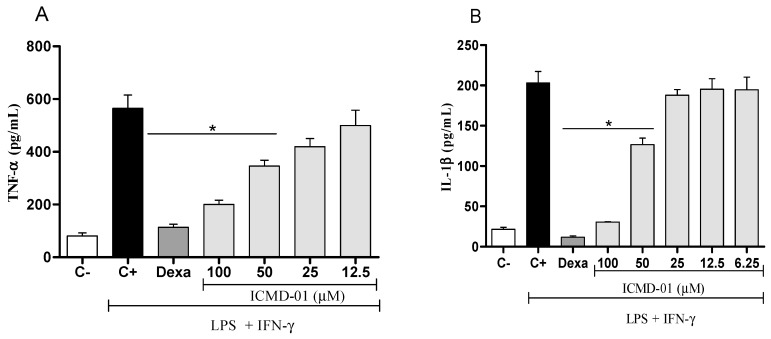
The effect of ICMD-01 on cytokine production by activated macrophages. Concentrations of TNF-α and IL-1β were determined in cultures of J774 macrophages (Panels (**A**) and (**B**)) treated with vehicle (5% DMSO in saline, control group), IMCD-01 (ICMD-01; 6.25, 12.5, 25, 50 or 100 μM) or dexamethasone (Dexa; 40 μM) in the presence of LPS (500 ng/mL) plus IFN-γ (5 ng/mL). Cell-free supernatants were collected 4 h (for TNF-α measurement) and 24 h (for IL-1β) after treatments for ELISA assays. Ct—shows cytokine concentrations in unstimulated cell. Data are expressed as means ± SEMs; *n* = 4 for TNF-α and *n* = 6 IL-1β determinations per group. * Significantly different from the vehicle treated cultures stimulated with LPS + IFN-γ (*p* < 0.05). ANOVA followed by Tukey´s multiple comparison test.

**Figure 4 ijms-21-02591-f004:**
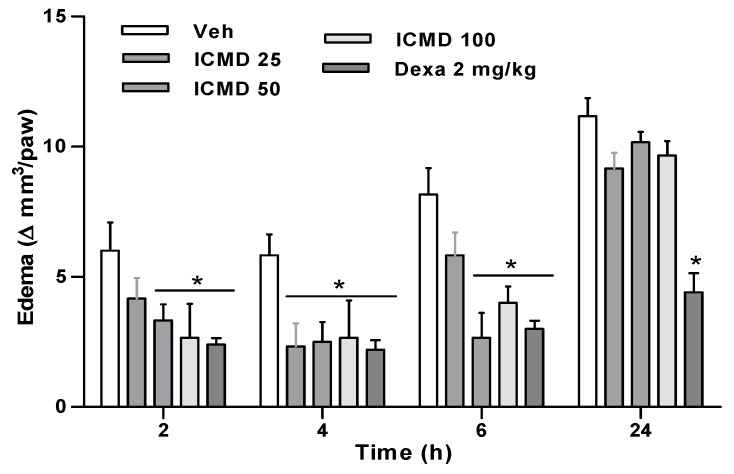
The effect of ICMD-01 on complete Freund’s adjuvant (CFA)-induced paw inflammation. Mice were injected with ICMD-01 (25, 50 and 100 mg kg^−1^), vehicle (5% DMSO in saline; control group) orally or dexamethasone (intraperitoneal route) (Dexa; 2 mg kg^−1^; reference drug) 40 min before CFA (injected at time zero). Paw edema was measured 2, 4, 6 and 24 h after CFA, represented as paw volume variation. Data are expressed as means ± SEMs; *n* = 6 mice per group. * Significantly different from the control group (*p* < 0.05). Two-way ANOVA followed by the Bonferroni’s test.

**Figure 5 ijms-21-02591-f005:**
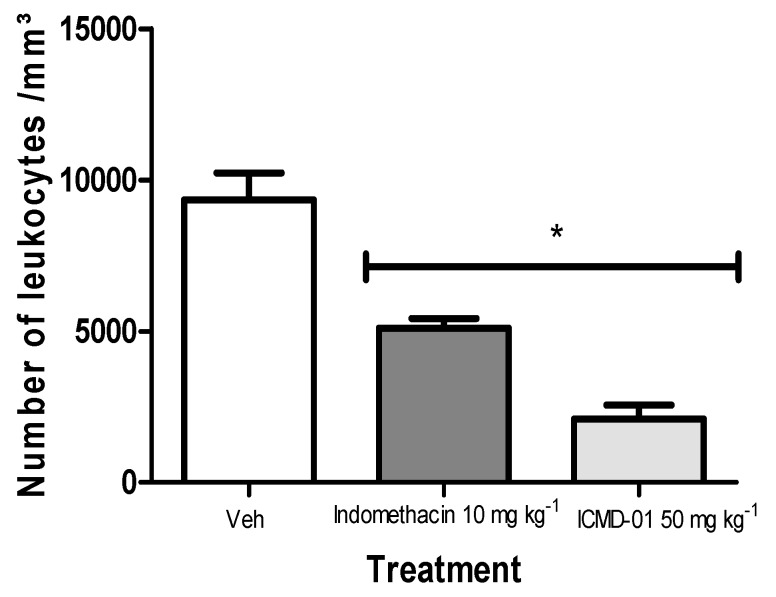
Effects of ICMD-01 on zymosan-induced peritonitis. Mice were injected with ICMD-01 (50 mg kg^−1^), vehicle (5% DMSO in saline; control group) or indomethacin (10 mg kg^−1^; reference drug) orally 30 min before zymosan 2%. The exudate was collected 4 h after induction, and is represented as a percentage of the number of leukocytes. Data are expressed as means ± SEMs; *n* = 5 mice per group. * Significantly different from the control group (*p* < 0.05). Two-way ANOVA followed by the Bonferroni’s test.

**Figure 6 ijms-21-02591-f006:**
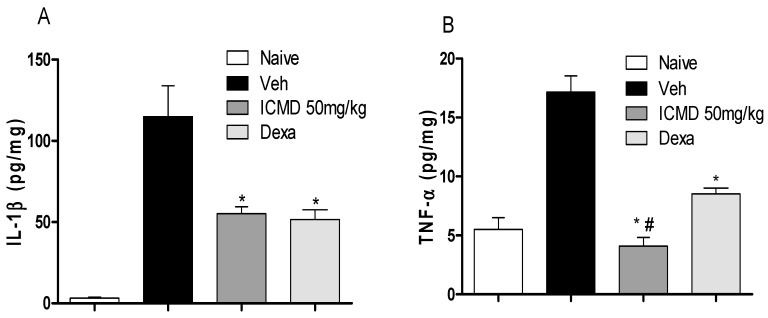
The effect of ICMD-01 on cytokine paw levels during CFA-induced inflammation. Mice were injected with ICMD-01 (50 mg kg^−1^), vehicle p.o. (5% DMSO in saline; control group) orally, or dexamethasone (intraperitoneal route) (Dexa; 2 mg kg^−1^; reference drug) route 40 min before CFA (injected at time zero). The naïve group consisted of mice that did not receive any experimental manipulation. Panels shows the paw levels of (**A**) interleukin-1β (IL-1β); (**B**) tumor necrosis factor-α (TNF-α), determined in skin tissues samples by ELISA, 3 h after the CFA injection. The results are expressed as picograms of cytokine per milligram of protein. Data are expressed as means ± SEMs; *n* = 6 mice per group. * Significantly different from the vehicle group in the same time (*p* < 0.05); ^#^ significantly different from the naive group (*p* < 0.05). ANOVA followed by Tukey´s multiple comparison test.

**Figure 7 ijms-21-02591-f007:**
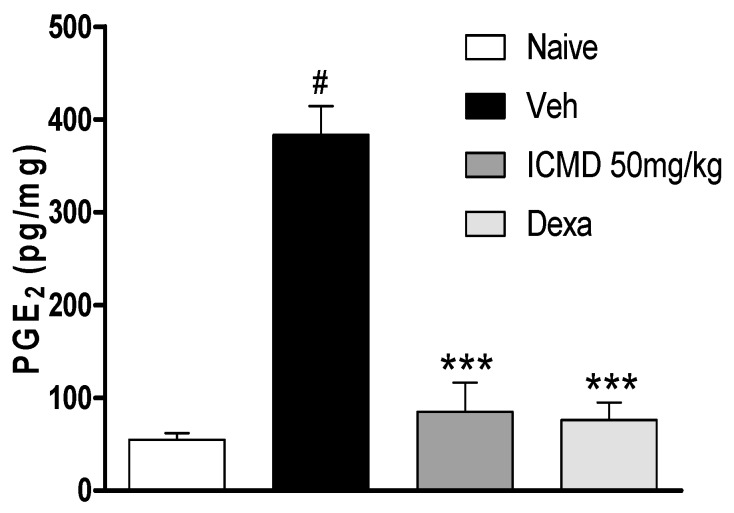
The effect of ICMD-01 on PGE_2_ paw levels during CFA-induced inflammation. Mice were treated orally with ICMD-01 (50 mg kg^−1^), dexamethasone (intraperitoneal route) (Dexa; 2 mg kg^−1^; reference drug), or vehicle (5% DMSO in saline; control group) 40 min before the intraplantar injection of CFA (20 μL/paw). The naïve group consisted of mice that did not receive any experimental manipulation. Panels show the paw levels of (prostaglandin E_2_ (PGE_2_), determined in skin tissues samples 3 h after the CFA injection. The results are expressed as picograms of PGE_2_ per milligram of protein. Data are expressed as means ± SEMs; *n* = 6 mice per group. *** Significantly different from the vehicle group in the same time (*p* < 0.05); ^#^ significantly different from the naive group (*p* < 0.05). ANOVA followed by Tukey´s multiple comparison test.

**Figure 8 ijms-21-02591-f008:**
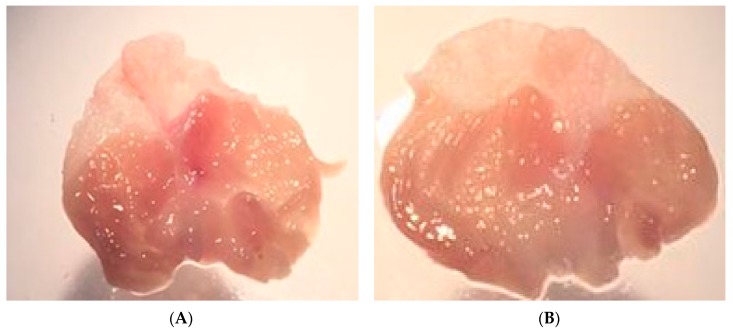
Stomachs of the control group (**A**) and the group treated with ICMD-01 (**B**). Mice were treated p.o. with ICMD-01 (50 mg kg^−1^) or vehicle (5% DMSO in saline; control group) for seven consecutive days.

**Table 1 ijms-21-02591-t001:** In silico pharmacokinetic data estimated with SwissADME or pkCSM web services. ^a^ SwissADME Moriguchi log of octanol-water partition coefficient. ^b^ SwissADME Ali log of aqueous solubility. ^c^ SwissADME calculation of Topological PolarSurface Area (TPSA). ^d^ pkCSM prediction of Caco-2 cell permeability estimation of absorption at human intestinal mucosa. ^e^ pkCSM prediction of the proportion of compound absorption though the human small intestine. ^f^ pkCSM prediction of the log of steady state volume of distribution (VDss). ^g^ pkCSM prediction of compound fraction unbound in plasma (not bound to serum proteins). ^h^ pkCSM prediction of the log of total drug clearance.

Compound	Log*P* ^a^	Log*S* ^b^	TPSA (Å^2^) ^c^	Caco-2 Permeability (log Papp; log cm/s) ^d^	Int. Abs. (%) ^e^	VDss (log L/kg) ^f^	Fract. Unb.^g^	Total Clearance (log ml/min/kg) ^h^
ICMD-01	2.40	−3.98	64.92	0.825	91.123	0.252	0.00	0.318

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
