# Peer review of "(E)-2-Cyano-3-(1H-Indol-3-yl)-N-Phenylacrylamide, a Hybrid Compound Derived from Indomethacin and Paracetamol: Design, Synthesis and Evaluation of the Anti-Inflammatory Potential"

_ijms, 2020, doi:10.3390/ijms21072591_

Round 1
Reviewer 1 Report
The authors report the design and anti-inflammatory activity of a novel hybrid compound containing features of both indomethacin and acetaminophen. Overall, the manuscript is decently-written and well-presented. The experiments appear to be competently designed and executed, and the conclusions are supported by the data. There are a few issues that need be addressed prior to publication:
- The description of the design of ICMD-01 is not consistent with how the molecule was actually discovered. FBDD typically involves screening molecular fragments for activity, with subsequent addition to fragment structure, and SBDD entails using structural aspects of the target to design molecules to bind. Neither one of these approaches were used here. Rather, the authors designed a molecule that contains structural elements of two known drugs to generate a hybrid molecule, and then used docking studies to see if there is potential for COX inhibition. There is nothing wrong with the approach the authors took; however, the manuscript needs to be modified to correctly describe how events leading to the discovery of ICMD-01.
- Given the efforts invested into modeling the binding of ICMD-01 to COX and the extensive subsequent pharmacological characterization, it is surprising that no in vitro COX inhibition assays were ran. This is a straight-forward assay that that really should be performed to contextualize the usefulness of this hybrid molecule.
- Inspection of the structure of ICMD-01 quickly reveals that this molecule is expected to be quite electrophilic. Gratifyingly, the authors discuss the possibility of Nrf2 activation as part of ICMD-01’s activity. Most Nrf2 activators are electrophiles/Michael acceptors. This aspect of ICMD-01’s chemistry should be added to the discussion.
- The structure of ICMD-01 in Figure 1 needs to be revised to ensure the cyano group is depicted in a manner that is consistent with its geometry.
- The current sentiment among medicinal chemists is that a molecule’s logP value is the most important of the Lipinski descriptors. A nice summary can be found here: Med. Chem. 2019, 62, 4, 1701-1714.
- Since only one molecule was prepared, the experimental section needs to be modified (i.e. the terms general procedure, yields, etc., should be changed).
Reviewer 2 Report
In this work, the authors show the hybrid (E)-2-cyano-3-(1H-indol-3-yl)-N-phenylacrylamide (ICMD-01) was designed using both fragment-based drug design (FBDD) and structure-based drug design techniques. The work is interesting and well written. However, the computational part has some gaps to fill. In my opinion, the paper might be considered to be accepted after major and minor revisions.
Major points
- Pages 2-3, lines 90-100: The discussion on docking procedure, showed in these sentences, is not clear: I suggest to organize the discussion better, starting from the energy comparison and explaining in detail similarity and differences in the binding mode of the synthesized compound with respect to indomethacin, explicitly referring also to the figures of the supplementary.
- Page 12, lines 405-406: “To validate the docking procedure, the co-crystalized ligand indomethacin was previously docked to each protein.” To validate the docking procedure, the authors must provide the RMSd value between the crystallographic pose and the one generated by the docking and they have to show the image in the supplementary.
Minor points
- Page 2, lines 83-85: “To validate possible anti-inflammatory actions, the study which the compound was anchored at the enzymatic site of cyclooxygenases was carried out.”. I think this sentence is not clear. Please, write the sentence again.
- Page 2, lines 85-87: “Moreover, indomethacin was used as a co-crystallized ligand for comparative purposes order to provide information on the possibilities of binding with less energy through its coupling with the receptor” Please, modify the sentence as follow: “Moreover, the co-crystallized ligand indomethacin was used as comparison in order to provide information on the possibility of a better binding energy respect to its coupling analogue with the receptor.”
- Page 3, Figure 1: The Figure is not clear, please modify. Moreover, I think the caption is not correct. Maybe the correct sentence is “3D and 2D interaction diagram of coupling ICMD-01with active site of target receptor human cyclooxygenase-1 (A) (3KKG) and cyclooxygenase-2 (B) (4COX).
- Page 11, lines 378-379: “In silico molecular interaction, analysis studies on the test compound with COX-1 and COX were performed using AutoDock4.2.6 in combination with the Lamarckian genetic algorithm.” Please, modify as follow: “In silico molecular interaction analysis studies on the new synthetized compound with COX-1 and COX-2 were performed using AutoDock4.2.6 in combination with the Lamarckian genetic algorithm.
- Page 11, lines 384-389: Review grammatical errors.
- Page 11, lines 391-392: “The crystal structures of indomethacin co-cristallized to COX-1 (PDB ID: 3KK6) and COX-2 392 (PDB ID: 4COX) were retrieved from RCSB Protein Data Bank as described.” The sentence is not entirely correct, because 3KK6 is the crystal structure of cyclooxygenase-1 in complex with celecoxib.
Round 2
Reviewer 1 Report
THe authors have satisfactorily addressed my comments with the one exception:
-The cyano group is sp hybridized, and as such, is always in a linear geometry. Please correct this in Figure 1.
Author Response
The cyan group does in fact show sp hybridization with linear geometry, however, for the study of molecular modeling, the image generated by the program is consistent with its availability in the most stable form for interaction with minimum energy value, and it is not possible to carry out this adjustment. Nonetheless, the image was generated again to make nitrile more evident in 3D images. Similar representations in docking studies containing the nitrile group can be found in the article Biorg Med Chem. 2018 Dec 1; 26 (22). 5911-5921.
Reviewer 2 Report
I'm glad that I could see the suggested changes,however in my opinion the figure 1 (especially 3D representations) remains unclear.
Author Response

(The authors gave the same response as above.)
